# Inpatient Burden and Mortality of Goodpasture’s Syndrome in the United States: Nationwide Inpatient Sample 2003–2014

**DOI:** 10.3390/jcm9020455

**Published:** 2020-02-06

**Authors:** Wisit Kaewput, Charat Thongprayoon, Boonphiphop Boonpheng, Patompong Ungprasert, Tarun Bathini, Api Chewcharat, Narat Srivali, Saraschandra Vallabhajosyula, Wisit Cheungpasitporn

**Affiliations:** 1Department of Military and Community Medicine, Phramongkutklao College of Medicine, Bangkok 10400, Thailand; 2Division of Nephrology, Department of Medicine, Mayo Clinic, Rochester, MN 55905, USA; charat.thongprayoon@gmail.com (C.T.); api.che@hotmail.com (A.C.); 3Department of Medicine, David Geffen School of Medicine, University of California, Los Angeles, Los Angeles, CA 90095, USA; boonpipop.b@gmail.com; 4Clinical Epidemiology Unit, Department of Research and Development, Faculty of Medicine, Siriraj Hospital, Mahidol University, Bangkok, 10700, Thailand; p.ungprasert@gmail.com; 5Department of Internal Medicine, University of Arizona, Tucson, AZ 85721, USA; tarunjacobb@gmail.com; 6Department of Internal Medicine, St. Agnes Hospital, Baltimore, MD 21229, USA; nsrivali@gmail.com; 7Department of Cardiovascular Medicine, Mayo Clinic, Rochester, MN 55905, USA; Vallabhajosyula.Saraschandra@mayo.edu; 8Division of Nephrology, Department of Medicine, University of Mississippi Medical Center, Jackson, MS 39216, USA; wcheungpasitporn@gmail.com

**Keywords:** Goodpasture syndrome, anti-GBM disease, epidemiology, hospitalization, outcomes

## Abstract

**Background:** Goodpasture’s syndrome is a rare, life-threatening, small vessel vasculitis. Given its rarity, data on its inpatient burden and resource utilization are lacking. We conducted this study aiming to assess inpatient prevalence, mortality, and resource utilization of Goodpasture’s syndrome in the United States. **Methods:** The 2003–2014 National Inpatient Sample was used to identify patients with a principal diagnosis of Goodpasture’s syndrome. The inpatient prevalence, clinical characteristics, in-hospital treatment, end-organ failure, mortality, length of hospital stay, and hospitalization cost were studied. Multivariable logistic regression was performed to identify independent factors associated with in-hospital mortality. **Results:** A total of 964 patients were admitted in hospital with Goodpasture’s syndrome as the principal diagnosis, accounting for an overall inpatient prevalence of Goodpasture’s syndrome among hospitalized patients in the United States of 10.3 cases per 1,000,000 admissions. The mean age of patients was 54 ± 21 years, and 47% were female; 52% required renal replacement therapy, whereas 39% received plasmapheresis during hospitalization. Furthermore, 78% had end-organ failure, with renal failure and respiratory failure being the two most common end-organ failures. The in-hospital mortality rate was 7.7 per 100 admissions. The factors associated with increased in-hospital mortality were age older than 70 years, sepsis, the development of respiratory failure, circulatory failure, renal failure, and liver failure, whereas the factors associated with decreased in-hospital mortality were more recent year of hospitalization and the use of therapeutic plasmapheresis. The median length of hospital stay was 10 days. The median hospitalization cost was $75,831. **Conclusion:** The inpatient prevalence of Goodpasture’s syndrome in the United States is 10.3 cases per 1,000,000 admissions. Hospitalization of patients with Goodpasture’s syndrome was associated with high hospital inpatient utilization and costs.

## 1. Introduction

Goodpasture’s syndrome (GS) is a rare, life-threatening, small vessel vasculitis that is mediated by circulating anti-glomerular basement membrane (anti-GBM) autoantibodies against the NC1 domain of the alpha 3 chain of type IV collagen, targeting capillaries of the kidneys and lungs [1,2,3,4]. It is considered as one of the organ-specific autoimmune diseases that typically presents as a rapidly progressive glomerulonephritis (RPGN), accompanied by alveolar hemorrhage with pathology characterized by crescentic glomerulonephritis with classic linear polyclonal immunoglobulin (Ig) G deposits on immunofluorescence staining of the GBM on analysis of kidney biopsy samples [1,2,5,6,7]. Without prompt diagnosis and treatment, patients with GS can develop organ failure, resulting in significant morbidities and mortality [1,2,3].

Among European and Asian populations, the incidence of GS is estimated to have a frequency of 0.5 to 1.8 cases per million population per year [1,2,3,8,9]. A recent study from Ireland reported a nationwide disease incidence of GS of 1.64 per million population per year [10]. While it is well known that patients with GS can have both pulmonary and renal involvement requiring a mechanical ventilator and renal replacement therapy [3,4], data on its inpatient burden and resource utilization are lacking.

Thus, we conducted this study using the 2003–2014 National Inpatient Sample (NIS) database to assess inpatient prevalence, mortality, and resource utilization of GS in the United States.

## 2. Materials and Methods

### 2.1. Data Source

The 2003–2014 NIS database was used to conduct this cohort study. The NIS is the publicly available, inpatient, all-payer database in the United States. This database was developed and maintained by the Healthcare Cost and Utilization Project (HCUP) under the sponsorship of the Agency for Healthcare Research and Quality (AHRQ). The dataset contains more than 7 million hospitalizations annually, which were obtained from a 20% stratified sample of over 4000 non-federal acute care hospitals in more than 40 states of the United States. A survey procedure using discharge weights provided by the HCUP-NIS database was used to generate national estimates for 95% of hospitalizations nationwide [11]. This dataset includes codes for principal and secondary diagnosis as well as codes for procedures performed during the hospitalization. 

### 2.2. Study Population

All patients with a principal diagnosis of GS, based on the International Classification of Diseases, Ninth Revision, Clinical Modification (ICD-9 CM) diagnosis code of 446.21 for the hospitalization were included.

### 2.3. Variables and Outcome of Interest

Patient characteristics included age, sex, race, year of hospitalization, smoking, hemoptysis, and the presence of anti-neutrophil cytoplasmic antibody (ANCA)-associated vasculitis, which consisted of granulomatosis polyangiitis, microscopic polyangiitis, and sepsis. Treatments included respiratory support consisting of invasive mechanical ventilation and non-invasive ventilation, renal replacement therapy, therapeutic plasmapheresis, and blood transfusion. Patient outcomes included organ failure or dysfunction, which consisted of respiratory failure, circulatory failure, renal failure, liver failure, hematologic failure, metabolic failure, and neurologic failure, as well as in-hospital mortality. Resource utilization included length of hospital stay and hospitalization cost.

### 2.4. Statistical Analysis

Discharge-level weights published by the HCUP were used to estimate the total number of GS patients. Continuous variables were summarized as mean ± standard deviation for normally-distributed data, and median with interquartile range for skewed data. Categorical variables were summarized as count with percentage. The annual inpatient prevalence of GS in hospitalized patients in the United States from 2003 to 2014 was calculated. Independent factors associated with in-hospital mortality were identified using multivariable logistic regression with the forward stepwise selection method. A two-tailed *p*-value of less than 0.05 was considered statistically significant. All analyses were performed using JMP statistical software (version 10, SAS Institute, Cary, NC).

## 3. Results

### 3.1. Patient Characteristics and In-Hospital Treatment

Of 93,377,054 hospital admissions during the study period, 964 patients were admitted to hospital with GS as the principal diagnosis. The mean age of patients was 54 ± 21 years; 47% were female, 65% were Caucasian, and 9% had a co-diagnosis of ANCA-associated vasculitis. Of patients with GS, 19% needed invasive mechanical ventilation, 5% needed non-invasive ventilation support, and 52% required renal replacement therapy. Plasmapheresis was performed in 39% of patients. Table 1 shows clinical characteristics and in-hospital treatment of GS patients in this cohort.

### 3.2. Inpatient Prevalence of GS

Table 2 shows the annual distribution and inpatient prevalence of GS in hospitalized patients. The inpatient prevalence of GS ranged from 6.7 to 12.1 per 1,000,000 admissions between the years 2003 and 2014 in the United States with an overall inpatient prevalence of GS over 12 years of 10.3 cases per 1,000,000 admissions (Figure 1).

### 3.3. Organ Failure and In-Hospital Mortality

Seventy-six percent of patients had at least one end-organ failure. Renal failure was the most common end-organ failure (62%), followed by respiratory failure (29%), metabolic failure (17%), hematologic failure (13%), circulatory failure (6%), neurological failure (5%), and liver failure (1%) (Table 1). The number of end-organ failures was significantly associated with increased in-hospital mortality with an adjusted OR of 2.19 (95% CI 0.45–10.58) for one end-organ failure, 7.60 (95% CI 1.67–34.56) for two end-organ failures, and 19.86 (95% CI 4.10–96.19) for ≥3 end-organ failures.

Of 964 patients with GS, 74 (8%) died in the hospital. In the multivariable logistic regression, age older than 70 years (OR 3.62; 95% CI 1.52–8.61 compared to age ≤ 39 years), sepsis (OR 5.38; 95% CI 2.53–11.45), respiratory failure (OR 7.41; 95% CI 3.85–14.26), circulatory failure (OR 7.85; 95% CI 3.37–18.26), renal failure (OR 2.55; 95% CI 1.21–5.37), and liver failure (OR 32.32; 95% CI 3.51–297.19) were associated with increased in-hospital mortality. In contrast, more recent year of hospitalization (OR 0.23; 95% CI 0.10–0.55 for year 2011–2014 compared to year 2003–2006) and the use of therapeutic plasmapheresis (OR 0.43; 95% CI 0.22–0.84) were associated with decreased in-hospital mortality (Table 3). 

### 3.4. Length of Hospital Stay and Hospitalization cost

The median length of hospital stay was 10 (IQR 5–18) days. The median hospitalization cost was $75,831 (IQR 31,687–163,201) (Table 1).

## 4. Discussion

To the best of our knowledge, our study is the first to evaluate inpatient prevalence, mortality, and resource utilization of GS in the United States. We demonstrated overall inpatient prevalence of GS among hospitalized patients in the United States of 10.3 cases per 1,000,000 admissions. The in-hospital mortality rate was 8%. The factors associated with increased in-hospital mortality were age older than 70 years, sepsis, the development of respiratory failure, circulatory failure, renal failure, and liver failure, whereas the factors associated with decreased in-hospital mortality were more recent year of hospitalization and the use of therapeutic plasmapheresis. 

GS is often described to have an incidence of GS of 0.5 to 1.8 cases per million population per year in European and Asian populations, primarily based on single-center biopsy or serology-based series [1,2,3,8,9,12]. A recent nationwide study from Ireland identified all GS cases over a decade via reference immunology laboratories and a nationwide pathology database over an 11-year period, which reported a disease rate of 1.64 per million population per year [10]. In this study, we utilized the United States inpatient hospitalization data from the NIS database and demonstrated inpatient prevalence of GS among hospitalized patients in the United States of 10.3 cases per 1,000,000 admissions. Although hospitalization for GS is infrequent, we found that hospitalized patients with GS commonly had high rates of end-organ failure, including renal failure (62%) and respiratory failure (29%). While 19% of hospitalization for GS required invasive mechanical ventilation, 52% required renal replacement therapy. The median hospitalization cost for GS was as high as $75,831.

Our findings confirmed a bimodal age distribution of GS, with younger patients <39 years having a male predominance, whereas older patients >60 years old were more frequently female [2,12]. We also observed an increase in the inpatient prevalence of GS from 2004 to 2007, which subsequently stabilized (Figure 1). Although the reason remains unclear, we speculated that this is because of the increasing awareness and widespread availability of diagnostic tests around that time [4,13]. Previous studies have suggested that, in addition to genetic factors, environmental factors can also trigger the development of GS, such as cigarette smoking, inhaled hydrocarbons, or potential infectious triggers damaging the alveolar basement membrane and exposing type IV collagen epitopes [3,4,6,14,15,16,17,18]. Future studies are needed to assess if these factors play an important role in the trends of inpatient prevalence of GS in the United States.

Our study demonstrated that 52% of hospitalization for GS required dialysis, which was consistent with previous literature indicating that approximately half of patients with GS require hemodialysis [19]. There are limited data on how frequently artificial ventilation is required. Small series estimated that this occurred in 11% of patients with GS [20,21,22,23]. Our study demonstrated that 19% of hospitalizations for GS needed invasive mechanical ventilation, and 5% needed non-invasive ventilation support. GS is life-threatening, with irreversible kidney damage and respiratory failure. Aggressive therapy with modern treatment protocols with antibody removal by plasmapheresis, use of corticosteroids, and immunosuppressive agents, particularly cyclophosphamide, has dramatically improved patient outcomes compared to the past [4,18,24,25,26]. The 5-year survival rate exceeds 80% and fewer than 30% of patients require long-term dialysis [2]. In our study, we demonstrated that in-hospital mortality rate of GS in the United States between the years 2003 and 2014 was 8%. Although the data on medication were limited in the database, we found that recent year of hospitalization and the use of therapeutic plasmapheresis were associated with decreased in-hospital mortality among patients with GS. While the underlying explanations of decreased in-hospital mortality among patients with GS in the recent years of hospitalization remain unclear and require further investigations, this finding may potentially represent an improvement in patient care of GS in recent years of hospitalization.

There are several limitations of this study. Firstly, although the utilization of the NIS database allowed us to evaluate U.S. inpatient prevalence and burden of patients with GS, possible inaccuracies in ICD-9 CM coding may have confounded the results. Secondly, given the administrative nature of the dataset, the data on medication such as immunosuppression were limited in this study. Consequently, we could not assess the potential effects of immunosuppression, such as cyclophosphamide treatment on hospital outcomes of patients with GS. Thirdly, this was an analysis of an inpatient database in the United States. Sixty-five percent of patient populations with GS in NIS database were Caucasian, and this limits generalizability to the patient population in other countries. Fourthly, kidney biopsy, and laboratory data were lacking in the database. Previous studies have suggested that no patient with 100% glomerular crescents and dialysis dependence at presentation recovered kidney function, and so current guidelines do not recommend treatment in these cases [27,28]. Furthermore, studies have also demonstrated that those patients with higher serum creatinine (5.7 mg/dL or higher) and reduced proportion of normal glomeruli on kidney biopsy have poor renal outcomes [2,13]. Therefore, future studies are needed to assess the impacts of kidney biopsy findings on the treatment and outcomes during hospitalizations for GS.

## 5. Conclusions

In summary, we demonstrate overall inpatient prevalence among patients with GS between the years 2003 and 2014 in the United States, with 10.3 cases per 1,000,000 admissions. Although the in-hospital mortality rate was only 8%, hospitalization of patients with GS was associated with high hospital inpatient utilization and costs.

## Figures and Tables

**Figure 1 jcm-09-00455-f001:**
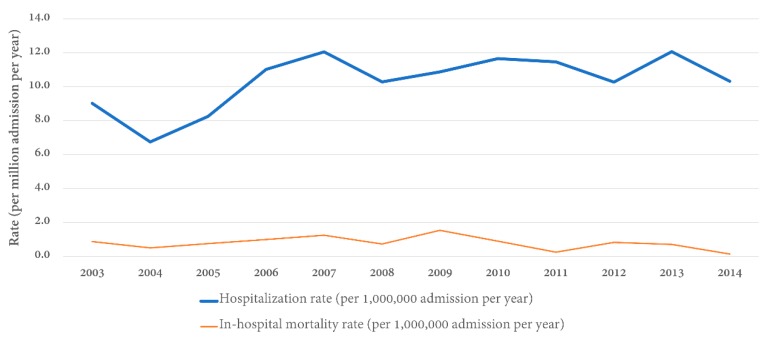
Rate of hospital admission and in-hospital mortality rate for Goodpasture’s syndrome stratified by year.

**Table 1 jcm-09-00455-t001:** Clinical characteristics, treatments, outcomes, and resource utilization of Goodpasture’s syndrome patients.

	All (*N* = 964)
**Clinical characteristics**	
Age (years)	54 ± 21
≤39	260 (27)
40–49	91 (9)
50–59	141 (15)
60–69	199 (21)
≥70	273 (28)
Male sex	456 (47)
Caucasian	622 (65)
Year of hospitalization	
2003–2006	281 (29)
2007–2010	357 (37)
2011–2014	326 (34)
Smoking	95 (10)
Hemoptysis	267 (28)
ANCA vasculitis	84 (9)
Granulomatosis with polyangiitis	54 (6)
Microscopic polyangiitis	30 (3)
Sepsis	62 (6)
**Treatments**	
Respiratory support	216 (23)
Invasive mechanical ventilation	181 (19)
Non-invasive ventilation	49 (5)
Renal replacement therapy	499 (52)
Hemodialysis	494 (51)
Peritoneal dialysis	10 (1)
Therapeutic plasmapheresis	376 (39)
Blood transfusion	391 (41)
**Outcomes**	
Number of organ failure	
0	230 (24)
1	369 (38)
2	242 (25)
≥3	123 (13)
Respiratory failure	283 (29)
Circulatory failure/shock	53 (6)
Renal failure	597 (62)
Liver failure	10 (1)
Hematologic failure	127 (13)
Metabolic failure	159 (17)
Neurological failure	50 (5)
In-hospital death	74 (8)
**Resource utilization**	
Length of stay (days), median (IQR)	10 (5–18)
<5	215 (22)
5–9	229 (24)
10–14	188 (20)
≥15	332 (34)
Hospitalization cost ($), median (IQR)	75,831.5 (31,687.3–163,201.0)

**Table 2 jcm-09-00455-t002:** The distribution and inpatient prevalence of Goodpasture’s syndrome from 2003 to 2014.

Year	Total Number of Goodpasture’s Syndrome Patients	Total Number of Admissions	Inpatient Prevalence (per 1,000,000 Admissions)
2003	72	7,977,728	9.0
2004	54	8,004,571	6.7
2005	66	7,995,048	8.3
2006	89	8,074,825	11.0
2007	97	8,043,415	12.1
2008	84	8,158,381	10.3
2009	85	7,810,762	10.9
2010	91	7,800,441	11.7
2011	92	8,023,590	11.5
2012	75	7,296,968	10.3
2013	86	7,119,563	12.1
2014	73	7,071,762	10.3
Total	964	93,377,054	10.3

**Table 3 jcm-09-00455-t003:** Univariable and multivariable analysis assessing factors associated with in-hospital mortality in Goodpasture’s syndrome patients.

Characteristics	Univariable Analysis	Multivariable Analysis
Crude OR (95% CI)	*P*-Value	Adjusted OR (95% CI)	*P*-Value
Age (years)		
≤39	1 (ref)		1 (ref)	
40–49	0.85 (0.23–3.17)	0.81	0.76 (0.17–3.35)	0.71
50–59	0.54 (0.15–2.01)	0.36	0.58 (0.13–2.52)	0.47
60–69	2.34 (1.05–5.22)	0.04	1.68 (0.62–4.54)	0.31
≥70	4.42 (2.16–9.02)	<0.001	3.62 (1.52–8.61)	<0.01
Male	1.34 (0.83–2.16)	0.23		
Caucasian	1.34 (0.83–2.17)	0.23		
Year of admission		
2003–2006	1 (ref)		1 (ref)	
2007–2010	1.11 (0.65–1.91)	0.70	0.92 (0.47–1.79)	0.80
2011–2014	0.46 (0.23–0.90)	0.02	0.23 (0.10–0.55)	0.001
Smoking	0.24 (0.06–0.99)	0.04		
Hemoptysis	1.99 (1.22–3.23)	<0.01		
Granulomatosis with polyangiitis	1.42 (0.59–3.43)	0.43		
Microscopic polyangiitis	0.41 (0.06–3.03)	0.38		
Sepsis	14.03 (7.85–25.10)	<0.001	5.38 (2.53–11.45)	<0.001
Respiratory failure	10.71 (6.04–19.02)	<0.001	7.41 (3.85–14.26)	<0.001
Circulatory failure/shock	11.72 (6.35–21.66)	<0.001	7.85 (3.37–18.26)	<0.001
Renal failure	3.10 (1.68–5.72)	<0.001	2.55 (1.21–5.37)	0.01
Liver failure	16.90 (3.71–76.99)	<0.001	32.32 (3.51–297.19)	<0.01
Hematologic failure	1.17 (0.60–2.28)	0.66		
Metabolic failure	1.32 (0.73–2.39)	0.36		
Neurological failure	3.32 (1.59–6.95)	0.001		
Invasive mechanical ventilation	11.26 (6.72–18.86)	<0.001		
Non-invasive ventilation	1.74 (0.71–4.23)	0.22		
Dialysis	1.40 (0.87–2.27)	0.17		
Therapeutic plasmapheresis	0.89 (0.54–1.46)	0.64	0.43 (0.22–0.84)	0.01
Blood transfusion	0.83 (0.51–1.36)	0.46

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
