# Peer review of "Inpatient Burden and Mortality of Goodpasture’s Syndrome in the United States: Nationwide Inpatient Sample 2003–2014"

_jcm, 2020, doi:10.3390/jcm9020455_

Round 1
Reviewer 1 Report
The authors clarified epidemiology and clinical course of Goodpasture’s syndrome in US, and the results were meaningful for the management of this disease. However, I cannot understand the system of National Inpatient Sample (NIS) database. Does it cover 95% hospitalization of whole of US, despite it was obtained from 20% stratified sample of over 4000 non-federal acute care hospitals? How did the authors treat patients transferred to a highly specialized hospital? Because Goodpasture’s Syndrome is rare disease, I guess several patients transferred a specialized hospital. If they counted both administration for one patient, data of in-hospital mortality could be an erroneous result.
The number of confounders in the multivariable logistic regression analysis was too much, because the number of deaths was limited (n=74). Please reconsider the confounders.
Please demonstrate the reason why one of the factors associated with decreased in-hospital mortality was more recent year of hospitalization.
Author Response
Reviewer: 1
Comment #1
The authors clarified epidemiology and clinical course of Goodpasture’s syndrome in US, and the results were meaningful for the management of this disease. However, I cannot understand the system of National Inpatient Sample (NIS) database. Does it cover 95% hospitalization of whole of US, despite it was obtained from 20% stratified sample of over 4000 non-federal acute care hospitals?
Response: We appreciate the reviewer’s important comment. The National Inpatient Sample (NIS) is the largest all‐payer database of hospital inpatient stays in the United States and contains discharge data from a 20% stratified sample of community hospitals. Survey procedures using discharge weights provided with HCUP‐NIS database were used to generate national estimates for >95% of all United States hospitals. We have additionally included this information in the method of our revised manuscript.
Reference: Introduction to the HCUP Nationwide Inpatient Sample 2009. http://www.hcup-us.ahrq.gov/db/nation/nis/NIS_2009_INTRODUCTION.pdf. Accessed February 01, 2020.
Comment #2
How did the authors treat patients transferred to a highly specialized hospital? Because Goodpasture’s Syndrome is rare disease, I guess several patients transferred a specialized hospital. If they counted both administration for one patient, data of in-hospital mortality could be an erroneous result.
Response: The reviewer raised very important point. In this study, we included only the first hospital admission primarily for Goodpasture’s syndrome for each patient and excluded hospital admission from another hospital transfer. Therefore, each hospital admission belonged to a unique patient. The hospital characteristics of included admission were shown in Table below.
|
Hospital region |
|
|
-Northeast |
15.1% |
|
-Midwest |
21.9% |
|
-South |
37.9% |
|
-West |
25.1% |
|
Hospital Local Teaching |
|
|
-Rural |
7.7% |
|
-Urban nonteaching |
34.3% |
|
-Urban teaching |
58.0% |
|
Hospital bed size |
|
|
-Small |
9.3% |
|
-Medium |
23.7% |
|
-Large |
67.0% |
Comment #3
The number of confounders in the multivariable logistic regression analysis was too much, because the number of deaths was limited (n=74). Please reconsider the confounders.
Response: The reviewer raises important point. We agree with the reviewer. Based on the rule of 10, the maximum number of factors that are reasonably included in multivariable logistic regression was about 8. When we re-examined factors in our multivariable model, hemoptysis and invasive mechanical ventilation could be represented by respiratory failure, and it was considered redundant to include all three factors in the model. Therefore, we decided to remove hemoptysis and mechanical ventilation from multivariable model. As the result, a total of 8 factors (including age, year of hospital admission, sepsis, respiratory failure, circulatory failure, renal failure, liver failure, and the use of plasmapheresis) were significantly associated with in-hospital mortality, as shown in Table below. We have updated our result of manuscript as reviewer’s suggestion.
|
Characteristics |
Univariable analysis |
Multivariable analysis |
||
|
Crude OR (95% CI) |
P-value |
Adjusted OR (95% CI) |
P-value |
|
|
Age (years) |
|
|
||
|
≤39 |
1 (ref) |
- |
1 (ref) |
- |
|
40-49 |
0.85 (0.23-3.17) |
0.81 |
0.76 (0.17-3.35) |
0.71 |
|
50-59 |
0.54 (0.15-2.01) |
0.36 |
0.58 (0.13-2.52) |
0.47 |
|
60-69 |
2.34 (1.05-5.22) |
0.04 |
1.68 (0.62-4.54) |
0.31 |
|
≥70 |
4.42 (2.16-9.02) |
<0.001 |
3.62 (1.52-8.61) |
<0.01 |
|
Male |
1.34 (0.83-2.16) |
0.23 |
- |
- |
|
Caucasian |
1.34 (0.83-2.17) |
0.23 |
- |
- |
|
Year of data collection |
|
|
||
|
2003-2006 |
1 (ref) |
- |
1 (ref) |
- |
|
2007-2010 |
1.11 (0.65-1.91) |
0.70 |
0.92 (0.47-1.79) |
0.80 |
|
2011-2014 |
0.46 (0.23-0.90) |
0.02 |
0.23 (0.10-0.55) |
0.001 |
|
Smoking |
0.24 (0.06-0.99) |
0.04 |
- |
- |
|
Hemoptysis |
1.99 (1.22-3.23) |
<0.01 |
- |
- |
|
Granulomatosis with polyangiitis |
1.42 (0.59-3.43) |
0.43 |
- |
- |
|
Microscopic polyangiitis |
0.41 (0.06-3.03) |
0.38 |
- |
- |
|
Sepsis |
14.03 (7.85-25.10) |
<0.001 |
5.38 (2.53-11.45) |
<0.001 |
|
Respiratory failure |
10.71 (6.04-19.02) |
<0.001 |
7.41 (3.85-14.26) |
<0.001 |
|
Circulatory failure/shock |
11.72 (6.35-21.66) |
<0.001 |
7.85 (3.37-18.26) |
<0.001 |
|
Renal failure |
3.10 (1.68-5.72) |
<0.001 |
2.55 (1.21-5.37) |
0.01 |
|
Liver failure |
16.90 (3.71-76.99) |
<0.001 |
32.32 (3.51-297.19) |
<0.01 |
|
Hematologic failure |
1.17 (0.60-2.28) |
0.66 |
- |
- |
|
Metabolic failure |
1.32 (0.73-2.39) |
0.36 |
- |
- |
|
Neurological failure |
3.32 (1.59-6.95) |
0.001 |
- |
- |
|
Invasive mechanical ventilation |
11.26 (6.72-18.86) |
<0.001 |
- |
- |
|
Non-invasive ventilation |
1.74 (0.71-4.23) |
0.22 |
- |
- |
|
Dialysis |
1.40 (0.87-2.27) |
0.17 |
- |
- |
|
Therapeutic plasmapheresis |
0.89 (0.54-1.46) |
0.64 |
0.43 (0.22-0.84) |
0.01 |
|
Blood transfusion |
0.83 (0.51-1.36) |
0.46 |
- |
- |
Comment #4
Please demonstrate the reason why one of the factors associated with decreased in-hospital mortality was more recent year of hospitalization.
Response: We agree with the reviewer regarding this important point. Thus, we have added this point in the discussion as the reviewer’s suggestion. The following Text has been added in the discussion of revised manuscript.
“Although the data on medication were limited in the database, we found that recent year of hospitalization and the use of therapeutic plasmapheresis were associated with decreased in-hospital mortality among patients with GS. While the underlying explanations of decreased in-hospital mortality among patients with GS in the recent years of hospitalization remain unclear and require further investigations, this finding may potentially represent an improvement in patient care of GS in recent years of hospitalization.”
We greatly appreciate the editor’s and reviewers’ time, expertise, and comments. They have significantly improved our manuscript.

Reviewer 2 Report
Introduction
Emphasize the importance of your research. What is the purpose of the study?
Material and methods
Specify the inclusion and exclusion criteria for the study more precisely. What about pharmacotherapy? What about chronic diseases?
Discussion / conclusions
The authors must further emphasize the usefulness of their research. What's new research findings for clinical practice?
Author Response
Reviewer: 2
Comment #1
Introduction
Emphasize the importance of your research. What is the purpose of the study?
Response: Thank you for reviewing our manuscript and for your critical evaluation. We sincerely appreciate your input and have found your suggestions very helpful.
We have revised introduction of our manuscript as suggested.
“Without prompt diagnosis and treatment, patients with GS can develop organ failures, resulting in significant morbidities and mortality [1-3].”
“While it is well known that patients with GS can have both pulmonary and renal involvement requiring mechanical ventilator and renal replacement therapy [3, 4], data on its inpatient burden and resource utilization are lacking.
Thus, we conducted this study using the 2003–2014 National Inpatient Sample (NIS) database to assess inpatient prevalence, mortality, and resource utilization of GS in the United States.”
Comment #2
Material and methods
Specify the inclusion and exclusion criteria for the study more precisely. What about pharmacotherapy?
Response: We appreciate the reviewer’s important point. We have additionally revised our material and methods as reviewer’s suggestion. There are several data that were limited including pharmacotherapy and we have included this information in the limitation of the study as suggested.
Comment #3
Discussion / conclusions
The authors must further emphasize the usefulness of their research. What's new research findings for clinical practice?
Response: The reviewer raises very important point. We agree and we have additionally emphasized the new findings of our study as reviewer’s suggestion.
“Our study is the first to evaluate inpatient prevalence, mortality, and resource utilization of GS in the United States. We demonstrated overall inpatient prevalence of GS among hospitalized patients in the United States of 10.3 cases per 1,000,000 admissions. The in-hospital mortality rate was 8%.”
“To the best of our knowledge, our study is the first to evaluate inpatient prevalence, mortality, and resource utilization of GS in the United States. We demonstrated overall inpatient prevalence of GS among hospitalized patients in the United States of 10.3 cases per 1,000,000 admissions. The in-hospital mortality rate was 8%. The factors associated with increased in-hospital mortality were age older than 70 years, sepsis, the development of respiratory failure, circulatory failure, renal failure, and liver failure, whereas the factors associated with decreased in-hospital mortality were more recent year of hospitalization, and the use of therapeutic plasmapheresis.”
All authors thank the Editors and reviewers for their valuable suggestions. The manuscript has been improved considerably by the suggested revisions!

Round 2
Reviewer 2 Report
Ready for publication